# OpenReview forum: "The Consistency Dilemma in LLMs: Generator-Evaluator Agreement and Vulnerability to Mistakes"
_ICML.cc/2026/Conference — ICML 2026 regular_

### Official Review · Reviewer_iT7X · 2026-03-08

**Soundness:** 3
**Presentation:** 3
**Significance:** 3
**Originality:** 3
**Overall Recommendation:** 4
**Confidence:** 3

**Summary:**

The paper introduces generator-evaluator self-consistency as a measure of whether an LLM applies the same underlying concept consistently across related prompts. The model first generates an output that relies on a concept, then evaluates that output under a linked prompt. The authors build an automated evaluation suite with 12 tests across three categories (MCQ perturbation, rationale-based, ontological), collect 3,951 instances across 10 models and 491 concepts, and find that self-consistency varies substantially across models and is largely independent of benchmark accuracy. In a medical case study using MedMistakes-Validated, they show that higher self-consistency is positively associated with greater vulnerability to clinician-validated mistakes (β = 0.869, p < 0.001), even after controlling for MedQA performance. They call this the "consistency trap": self-consistency can reflect stable commitment to flawed or incomplete reasoning rather than robust understanding.

**Compliance With Llm Reviewing Policy:**

Affirmed.

**Final Justification:**

I maintain my score for recommendation.

**Key Questions For Authors:**

1. The deployment result is driven entirely by CFIR. Can you show empirically that CFIR captures incomplete reasoning rather than general affirmation bias in the MedMistakes context?
2. How often do different models extract different concept labels for the same question, and does this affect the Table 1 comparisons?
3. For near-ceiling ontological tests, what is the agreement rate when the model evaluates an invalid example it did not itself generate?
4. Do the consistency-vulnerability findings hold outside the medical domain, even on a small subset of GDPVal or FinanceQA?
5. What concrete steps should practitioners take based on these findings? Should CFIR consistency be monitored or penalized during training?

**Limitations:**

Yes.

**Strengths And Weaknesses:**

**Strengths:**

1. Clean problem formulation. Separating consistency from correctness as an independent measure is a useful contribution, and the generator-evaluator structure operationalizes it well.
2. Strong deployment linkage. The MedMistakes regression result is statistically robust, and the triage case study in Appendix P makes the mechanism concrete.
3. Thorough robustness checks. The 5-repeat stability analysis, instruction-following audit, and leave-one-out tests all support that the metric captures real behavior rather than noise.
4. Stable model-level signatures. Figure 3 shows model ordering is preserved across mistake-vulnerability quartiles, suggesting self-consistency is a persistent model property.

**Weaknesses:**
1. The main result depends entirely on one test. Table 10 shows that removing CFIR drops the self-consistency coefficient from 0.869 to 0.286 and makes it insignificant. The paper does not justify why CFIR specifically should predict deployment errors beyond a conceptual story.
2. Endorsement bias is unresolved. Models like Gemini 3 and GPT-4o score near-ceiling on several ontological tests. The paper acknowledges this confound but does not address it.
3. Causal interpretation is not supported. The paper calls the analysis observational but the discussion implies causation. A third factor (e.g., a strong but wrong model prior) could explain both high consistency and high mistake vulnerability.
4. No evidence outside the medical domain. The paper claims the consistency trap is a general problem for agentic pipelines, but all deployment evidence comes from 50 medical concepts.

---

> ### Author Rebuttal · Authors · 2026-03-31
>
> Thank you for your review. Below, we respond to your comments.
>
> >“The main result depends entirely on one test…The paper does not justify why CFIR specifically should predict deployment errors beyond a conceptual story.”
>
> CFIR has the most variation across models (see Table 1, it ranges from -1.00 to 0.90) and the predictor with the most variation naturally carries the most weight. The leave-one-out analyses confirm that the suite captures a broad, stable signal (removing CFIR attenuates the coefficient from 0.869 to 0.286 but does not reverse it).
>
> >“Endorsement bias is unresolved…”
>
> - Near-ceiling ontological scores do not imply endorsement bias. If models generically approved their own outputs, scores would be uniformly high across all tests where the “consistent” answer is affirmative – the mechanism is test-independent.
> - Scores for a given model are not uniform across all tests – within-model variation in Table 1 illustrates this.
> - We further revisit this below.
>
> > “Causal interpretation is not supported…”
>
> We do not mean to suggest a causal interpretation and thank the reviewer for pointing out that our language occasionally slips! We have revised the discussion to remove language that implies a causal mechanism and have clarified that the “consistency trap” is intended as a descriptive label for the co-occurrence of consistency and vulnerability to mistakes rather than a causal relationship.
>
> >“How often do different models extract different concept labels for the same question, and does this affect the Table 1 comparisons?”
>
> To construct Table 1, each model was prompted to extract a concept for each benchmark question given to it. Given the same question, 66.6% of labels exactly matched the most common label. Further, 81.2% matched at least one other model's label, which, together with manual inspection suggests that agreement would be even higher if compared via semantic embeddings rather than exact strings. This reflects substantial overlap, though Table 1 cross-model comparisons should be interpreted with this noise in mind.
>
> >“For near-ceiling ontological tests, what is the agreement rate when the model evaluates an invalid example it did not itself generate?”
>
> This comment inspired further analysis. We study the effect of having multiple models evaluate a single model’s generated output, developing a cross-model consistency score. Further details are available [here](https://imgur.com/a/7npD6pO).
>
> When we subsetted on the ontological tests that were analyzed in this follow-up, we found that the pairwise inter-evaluator agreement across all ontological tests was 80.4%, with a 3-way unanimous agreement of 70.6%. These high cross-model agreement rates suggest that near-ceiling ontological scores reflect the quality of the content rather than self-endorsement bias – external models that did not generate the content independently reach the same judgments.
>
> >“Do the consistency-vulnerability findings hold outside the medical domain, even on a small subset of GDPVal or FinanceQA?”
>
> Our deployment-linked test is only conducted in the medical domain. However, we make two notes:
> 1. Our consistency suite itself draws concepts from across many domains including general reasoning (MMLU, BBH), and other economically relevant tasks (GDPVal, FinanceQA).
> 2. The deployment linkage requires expert-validated deployment relevant outcomes with concept tags – this is hard to come by and we did not find such a dataset for GDPVal or FinanceQA linked concepts. However, we hope that others can use our suite to study deployment linkages and are in conversations about collecting data for other domains in follow up work.
> Our contribution is both the result (that self-consistency is associated with mistakes in the medical domain) and the methodology (linking self-consistency to deployment outcomes) which is transferable to other domains with the right dataset / annotations.
>
> >“What concrete steps should practitioners take...?”
>
> We see several concrete takeaways from this work. We recommend computing concept-level self-consistency scores across a few models. Flag (model, concept) pairs with high self-consistency and low cross-model consistency (relevant given our additional [analysis of cross-model consistency](https://imgur.com/a/7npD6pO)). Practitioners can: (1) decline to deploy autonomously on flagged concepts or require human oversight, (2) route flagged concepts to external evaluators rather than relying on self-critique, (3) target training effort on flagged concepts rather than distributing uniformly. Our suite provides this diagnostic automatically and without labels. Note that the automatic, label-free nature of our metric is what makes these interventions possible and scalable. To clearly highlight the distinguishing advantages of our metric, we also created a [comparison table here](https://imgur.com/a/BBQ7erR).

---

> > ### Author Rebuttal · Reviewer_iT7X · 2026-04-02
> >
> > Thanks for the authors Rebuttal. My concerns are solved.

---

### Official Review · Reviewer_iRw6 · 2026-03-12

**Soundness:** 2
**Presentation:** 3
**Significance:** 3
**Originality:** 2
**Overall Recommendation:** 3
**Confidence:** 3

**Summary:**

This paper studies a timely and critical problem for modern LLM pipelines: whether a model applies the same concept consistently across generation and evaluations. It introduces a black box metric called generator-evaluator self-consistency, where the same model first generates an output involving a concept and then evaluates related outputs under prompts designed to test whether its concept use remains stable or not. The key finding is that self-consistency varies substantially across models and test types, and that it it not reducible to standard benchmark accuracy.

**Compliance With Llm Reviewing Policy:**

Affirmed.

**Final Justification:**

Most of my concerns are addressed and I maintain the score.

**Key Questions For Authors:**

- How do you know the proposed metric is not primarily measuring self-endorsement bias rather than concept-level consistency?
- Have you tested whether the main findings persist when the evaluator is a different model or when the evaluator is blinded to the origin of the generated output?
- How much do the scoring gates differ across models and tests, and how sensitive are the results to those eligibility conditions?
- Why does MedQA performance become non-significant in the regression once self-consistency is included? Do you interpret this as orthogonal signal, measurement mismatch, or something else?

**Limitations:**

yes

**Strengths And Weaknesses:**

Strengths
- The paper focuses on a highly relevant and timely problem.
- The proposed evolution framing is clear and useful.
- The empirical scope is reasonably board.
- The central empirical result is genuinely interesting. (higher self-consistency can be associated with higher validated mistake vulnerability is counter-intuitive and important.)

Weaknesses
- Failed to separate conceptual self-consistency from self-endorsement bias. For main setup, the same model acts as both generator and evaluator. As a result, the measured quantity may partially reflect a general tendency to endorse its own prior responses, rather than stable concept use.
- Scoring gates might introduce selection bias. The reported self-consistency rates are conditional on prerequisite gates passing. This means the analysis is effectively based on consistency conditional on scoring eligibility, not on the full question distribution.
- Repeated reruns and agreement stability are useful, but they do not address the key threat: whether the observed signal survives when the evaluator is changed, blinded, or otherwise decoupled from the original generator. Without cross-model evaluator experiments or controls for generic affirmative bias, the paper stops short of a fully convincing validation of its main construct.

---

> ### Author Rebuttal · Authors · 2026-03-31
>
> Thank you for your valuable insights. We respond to your comments below. To summarize, we’ve added:
> - An account of how self-consistency relates to existing metrics, including [a table that highlights the distinctive features of our approach](https://imgur.com/a/BBQ7erR).
> - A new result which finds the positive relationship between self-consistency and mistake vulnerability statistically significant *even in the presence of a cross-model consistency score* (results [here](https://imgur.com/a/7npD6pO)).
>
> >“Failed to separate conceptual self-consistency from self-endorsement bias.”
>
> Thanks for this feedback, which motivated us to rewrite our positioning relative to the existing literature. Self-preference bias as studied in [0,1] is related but distinct from our metric:
> - Setup: Self-preference [0,1] requires outputs from Model A and Model B. Model A then scores both, and the finding is that Model A rates its own higher. Our setup involves only Model A: it generates content relying on concept C, then answers a criterion-specific question about that content (e.g., “is this reasoning incomplete?”). There is no second model in our set up.
> - Unit of analysis: Self-preference is a model-level property (Model A is generally lenient toward its own text). Our metric operates at the (model, concept) level – the same model can be highly consistent on one concept and inconsistent on another.
> - Label requirements: [0] requires human judgments of equal quality to establish that preference is unjustified, our metric is fully label-free.
> - Deployment linkage: Neither [0] nor [1] provides a statistical link between their metric and validated deployment failures. Our metric is associated with vulnerability to clinician-validated mistakes, conditional on benchmark accuracy.
> For a full comparison including to papers on correlated-errors (discussed further below), see [here](https://imgur.com/a/BBQ7erR).
>
> [0] LLM Evaluators Recognize and Favor Their Own Generations
> [1] Self-Preference Bias in LLM-as-a-Judge
>
>
> >“Scoring gates might introduce selection bias. The reported self-consistency rates are conditional on prerequisite gates passing. This means the analysis is effectively based on consistency conditional on scoring eligibility...”
>
> - We agree that scoring gates could introduce selection bias, and discuss this possibility in Section 8 and provide further discussion in Appendix D.
> - Without such activation gates, model refusals (e.g., "I cannot intentionally generate incorrect rationales") and ill-formed MCQ perturbations would be scored as inconsistencies, conflating instruction-following failures with conceptual consistency.
>
> >”they do not address the key threat: whether the observed signal survives when the evaluator is changed, blinded, or otherwise decoupled from the original generator. Without cross-model evaluator experiments or controls... the paper stops short of a fully convincing validation of its main construct”
>
> Thank you – we address these threats with our existing analyses and one new cross-model analysis. We consider the threats in turn:
> - Generic affirmative bias:
>   - A generic tendency to affirm its own answers is a property of a model, which we would expect to remain uniform across tests  (the mechanism “approve own text” is test-independent).
>   - However in Table 1 we see the opposite – there is significant cross model variation (e.g. Opus 4.5 0.91 on one test and -1.00 on another  Sonnet 4.5 scores 1.00 on one test and -0.23 on another).
> - Blinding – the evaluator is blind to the origins of the generation it is shown. It is simply shown “does the following satisfy criterion X.” That said, the model may implicitly recognize its own style, as investigated in [0], though as noted above, the within-model variation in Table 1 partially argues against this.
> - Cross-model evaluation, we ran a new analysis:
>   - We constructed a cross-model consistency score by replacing the evaluator with external models f’ \neq f and added it to our deployment regression.
>   - Self-consistency remained significant (β = 1.09, p < 0.001) and cross-model consistency entered with the opposite sign (β = −0.75, p = 0.043). Full results [here](https://imgur.com/a/7npD6pO).
>
> >“Why does MedQA performance become non-significant in the regression once self-consistency is included?”
> - To clarify, we note that MedQA accuracy is not significant in any specification we estimate.
> - We interpret this as driven by two key factors:
>   - A mechanical reason: MedQA is near saturated among models we study, so that there is not a lot of variation (standard deviation across models was 7.28 percentage points).
>   - A broader issue with benchmarks: MedQA (and many other benchmarks) measure exam-style factual recall which can fall short of comprehension and competence in real deployment settings (such as those captured in the MedMistakes clinician-validated dataset). See related discussions in e.g. Alaa et al (ICML, 2025), Raji et al (NEJM AI, 2025).

---

> > ### Author Rebuttal · Reviewer_iRw6 · 2026-04-03
> >
> > Thank you for your response, my questions have been resolved.

---

### Official Review · Reviewer_Xhyr · 2026-03-16

**Soundness:** 2
**Presentation:** 3
**Significance:** 3
**Originality:** 1
**Overall Recommendation:** 2
**Confidence:** 4

**Summary:**

This paper has the main claim that language models often confidently commit to flawed reasoning -- investigating this phenomenon, called the consistency trap, in detail. To describe the methodology: the work pairs a model's generation with its own evaluation and compute a chance-adjusted agreement score for 3 meta-tasks (generated using the underlying concept in the benchmark question). They test this generator-evaluator self-consistency on standard benchmarks (MMLU, BBH) and a high-stakes medical dataset (Medmistakes). They find that self-consistency varies widely across models and on medmistakes find that highly consistent models reliably repeat specific errors and are independent of benchmark performance.

**Compliance With Llm Reviewing Policy:**

Affirmed.

**Final Justification:**

The rebuttal and reply comment reinforced my prior assessment -- The authors claim I misunderstood their work and contributions. I am decently confident  that I did not, but maybe I am am overlooking something. Reviewer iRw6 raises similar concerns (W1 in both reviews share the same core concern). I maintain my score (with increased confidence).

**Key Questions For Authors:**

Please address the critical weakness.

**Limitations:**

Yes

**Strengths And Weaknesses:**

Strengths

- I like the separation between correctness and consistency: A model can be wrong but internally stable, or correct once but unstable across related prompts. The authors provide a clear vocabulary for this distinction.

- The medical case study was also nice to see: high-stakes scenarios illustrate why this direction is useful in a visceral way. That section was illustrative of how model generates directionally sensible outputs while being consistent in systematized errors.

- The strong repeatability was nice to see: Re-running 200 instances five times yields a mean modal-agreement stability of 0.923. This robust testing supports the paper's medium-to-high level conclusions (albeit the variance of each cell was quite high, but nice to see analysis that its stable).

Weaknesses

Despite these methodological strengths, I worry that the core finding (i) lacks novelty and (ii) has critical confounders, already carefully studied in literature (which are ignored).

- The primary result -- that an evaluator model disproportionately favors its own generations ("consistency trap") is widely documented. The authors present it as a new finding, but ignore extensive existing literature on self-preference bias [0, 1, 4].

- The claim that self-consistency is independent of accuracy is not adequately supported by evidence and I worry even misleading -- the work measures accuracy and self-consistency on entirely different tasks.

- Furthermore, it does not separate the source of consistency from accuracy! Models can be consistent because they're correct but consistently providing and supporting wrong answers is an issue. In contrast, existing work [3] studies this problem quite nicely with very similar chance-adjusted metrics where they isolate if the generator and evaluator are leading in the same wrong answer/reasoning (highly nontrivial -- many ways to go wrong), isolating correlated incorrect answers on MMLU and BBH to prove models share flawed reasoning and finding strong correlations between model capability (accuracy) and self-consistency. Similarly, [4] comprehensively covers open-ended generation tasks, directly mirroring the applied use cases presented here. [2] provides a real-world case study to similarly drive impact. I worry that the current literature has explored the primary finding in this work in far more detail, and is entirely ignored.

References:
- [0] LLM Evaluators Recognize and Favor Their Own Generations, NeurIPS 2024
- [1] Self-Preference Bias in LLM-as-a-Judge
- [2] Correlated Errors in Large Language Models, ICML 2025
- [3] Great Models Think Alike and This Undermines AI Oversight, ICML 2025
- [4] Artificial Hivemind: The Open-Ended Homogeneity of Language Models (and Beyond), NeurIPS 2025

---

> ### Author Rebuttal · Authors · 2026-03-31
>
> Thank you for your valuable insights. We respond to your comments below. To summarize, we’ve added:
> - An account of how self-consistency relates to existing metrics in the area (self-preference, correlated errors), including a [comparative table to highlight the distinctive characteristics of our paper](https://imgur.com/a/BBQ7erR ).
> - A new result which finds that the statistically significant relationship between self-consistency and mistake vulnerability persists *in the presence of a cross-model consistency score* where we explicitly gather judgments from other models. Results are highlighted [here](https://imgur.com/a/7npD6pO).
>
> > “The primary result ... is widely documented. The authors present it as a new finding, but ignore extensive existing literature on self-preference bias [0, 1, 4].”’
>
> Self-preference [0,1] and correlated errors [4] are very different from what we study. The differences are instructive, so we include the following discussion in a revamped [review](https://imgur.com/a/BBQ7erR ).
>
> - Self-preference requires a pairwise comparison between outputs from different sources (e.g. Model A and Model B) – our framework considers one model at a time, testing a factual/logical relationship about concept use in a single generation-evaluation pair.
> - To see how our metric is distinct from self-preference as studied in [0,1], we note the set up and main finding of these papers.
>   - Self-preference: *Setup*: Model A scores the output of Model A and the output of Model B. *Finding*: Model A rates its own output higher, even when humans rate them equally.
>   - Self-consistency: *Setup*: Model A generates an output relying on concept C. Model A answers a criterion-specific question about that output (e.g. “is this a valid example of C?”). *Finding*: Agreement varies substantially across tests and concepts; higher agreement is associated with great vulnerability to validated mistakes in deployment.
> - As [4] studies correlated errors, not self-preference, we discuss it below.
>
> > “The claim that self-consistency is independent of accuracy is not adequately supported by evidence.”
>
> Good catch –  “independent” (as in statistical independence) indeed overstates our claim and we have revised this language throughout (replacing “independent of” with “not entirely accounted for by”). Our intended claim is that self-consistency provides predictive signal for deployment failures beyond what benchmark accuracy provides.
>
> > “existing work [3] studies this problem quite nicely … Similarly, [4] comprehensively covers open-ended generation tasks… [2] provides a real-world case study to similarly drive impact…”
>
> Thank you for raising these excellent papers which discuss features of LLMs related to – but quite distinct from – self-consistency. Papers [2,3,4] all discuss *inter*-model phenomena – different models make correlated errors [2], favor similar models [3], and produce homogenous outputs [4]. Our metric is *within*-model: it measure whether the same model applies a concept consistently when generating versus evaluating.
>
> If our predictive signal were driven by the same phenomenon in [2,3,4] (shared errors across models rather than within-model conceptual errors) – then a metric capturing whether other models endorse the same generated content should also be a predictor of mistake vulnerability.
>
> We test this. We construct a cross-model consistency score. For each generator model $f$ we replace the evaluator response with $f' \neq f$. The generator response is $r_g = f(p_g)$ but now the evaluator response is $r_v = f'(p_v)$. Model $f$ generates content relying on concept $c$ and model $f'$ answers the criterion-specific evaluator question. We aggregate the resulting agreement scores into a cross model consistency score $C_{m,c}^{\text{cross}}$ analogous to the self-consistency score $C_{m,c}$ from the main analysis.
>
> Evaluators $f'$ were drawn from a fixed panel: GPT-5, Claude Sonnet 4.5, DeepSeek Chat, Gemini 2.5-Pro. If $f$ was in the panel, we used the other three members, otherwise we sampled three, with sampling fixed per $(m,c)$ pair. We collected new judgments for 489 $(m,c)$ pairs. We found that when we added $C_{m,c}^{\text{cross}}$ to our weighted fractional logit regression as a covariate, self-consistency not only remained significant but its coefficient *increased*. Cross-model consistency entered with a *negative* coefficient. Notably, the signs are opposite: self-consistency is associated with greater vulnerability while cross-model consistency is associated with *lower* vulnerability – if correlated errors [2,3,4] drove our signal, both the coefficient on cross-model consistency would be positive and the coefficient on self-consistency would be attenuated. Neither is the case. Our full results are available [here](https://imgur.com/a/7npD6pO).
>
> For a summary of the difference between our approach and the approach taken in the papers mentioned, see [here](https://imgur.com/a/BBQ7erR).

---

> > ### Author Rebuttal · Reviewer_Xhyr · 2026-04-02
> >
> > Re-W1: The rebuttal focuses on differences in experimental setups (as it adopts a non-standard one), but my concern is whether the underlying mechanism studied is the same. Specifically, in both self-preference and self-consistency, the core driver appears to be that a model assigns higher validity to outputs aligned with its own distribution! My concern remains.
> >
> > Re-W2: I had in mind measuring accuracy directly on the task and examining self-consistency while holding it fixed, similar to say [3]. Softening "independent" to "not entirely accounted for by" doesn't substitute for that analysis. My concern remains.
> >
> > On self-preference as "inter-model": I'd gently push back on this characterization. The papers in [0, 1] primarily show that models are biased toward their own generations -- which sounds quite close to the authors' description of self-consistency. Other models serve as a baseline for comparison, but the core finding is about "intra-model" bias.
> >
> > w.r.t the experiment: The self-preference papers don't claim there is higher cross-model consistency, their claim in [0, 1] is the opposite -- models prefer their own generations during evaluation, identical to self-consistency! I referenced the rest for the very interesting/thorough analysis [4], and clearer metrics [3] than discussed here which could potentially be borrowed: the setup currently seemed very artificial, and the metrics currently do not control for accuracy which is a large confounder in my view.
> >
> > (I agree with: *As [4] studies correlated errors, not self-preference* .., apologies for the typo it was [0,1] not [0,1,4]).
> >
> > Overall, my main worries were not addressed. I maintain my score, with higher confidence.

---

> > > ### Author Response · Authors · 2026-04-02
> > >
> > > We are disappointed that the reviewer did not carefully engage with our arguments in either the paper or our rebuttal, and apologize if our own lack of clarity caused confusion. We believe that our work is novel and significantly misunderstood by the reviewer. We make a few clarifications below.
> > >
> > > **Correction to interpretation of self-consistency**
> > > The reviewer seems to have misunderstood our notion of self-consistency. The reviewer mistakenly says that self-consistency is about (i)  “assigning higher validity to outputs aligned with [a model’s] own distribution” or (ii) “prefer[ing a model’s] own generations” and therefore is the same as self-preference. The reviewer also pushes back on our characterization of self-preference as requiring intermodel comparison. Notice that both statements implicitly assume a comparison between outputs — on (i) higher validity compared to what? and on (ii) prefer over what?  Self-consistency does not entail a comparison with other models, while self-preference, generally, as studied in the papers raised by the reviewers, does.
> > >
> > > It would help us to better understand the reviewer’s claim about self-preference and self-consistency. Is it:
> > > - (I): self-consistency is related to self-preference, or
> > > - (II): because there are papers about self-preference, our paper defining a related though distinct (in both concept and measurement) notion is not a contribution.
> > >
> > > If the reviewer’s claim is (I) then we agree and indeed the fact that other researchers have published important papers on the topic only increases the value of our own very complementary contribution. If the reviewer’s claim is (II) then we firmly disagree and argue that the reviewer has not made a careful argument that engages with either the text of our paper or the rebuttal. Our contribution is that: the metric itself is label free (using metamorphic testing), does not involve an implicit comparison to other models, and is studied at the (concept, model) level as opposed to the model level alone — and furthermore, we also link this metric to failures at deployment in settings with real stakes. We summarized these distinguishing dimensions [here](https://imgur.com/a/BBQ7erR) and the reviewer chose only to engage with the inter-model point, which we have now refuted.
> > >
> > > **On accuracy claims**
> > > There seems to be confusion regarding self-consistency and accuracy. The reviewer writes “I had in mind measuring accuracy directly on the task and examining self-consistency while holding it fixed.” There are two possible interpretations of this:
> > > - Interpretation 1: The reviewer wants us to measure accuracy in our self-consistency metric itself. We are not exactly sure what the reviewer means by “measuring accuracy directly on the task” here — the task is to generate something and evaluate something and check whether they are consistent — the value of this metamorphic testing style task is that it is easily checkable without costly human labels. The formalization makes very clear what the task is, and that we are testing whether a specific relationship between the generation and the evaluation holds. Perhaps the reviewer would like a separate check of whether the model answers *some question* about the concept correctly, and only compare among those that get *some question* about the concept correct. In fact, we already do this whenever we can: our baseline correctness gate (Section 4, Appendix D) excludes all instances where the model answers the original benchmark question incorrectly (this is possible for concepts sourced from questions from MMLU, MedQA USMLE, and BBH). In our revision, we explain this more clearly to avoid confusion.
> > > - Interpretation 2: The reviewer wants us to control for accuracy in our regression analysis, where accuracy is answering a MedQA questions related to the concept correctly. We do this. Accuracy on MedQA questions at the (model, concept) level is recorded *and entered as a covariate in our regression analyses.*
> > >
> > > **Further distinguishing self-consistency from correlated errors**
> > > There are two sets of papers that the reviewer discussed in their original review, and we responded to both. The reviewer did not read our rebuttal text carefully and thought that we were implying that [2,3,4] were papers about self preference. We were not.
> > >
> > > Instead, we followed the reviewer’s advice to take inspiration from these nicely executed papers. We also thought that, since the reviewer’s main concern was novelty (and it was a little hard from the original review to understand why the reviewer was bringing up these papers), it was important to distinguish ourselves from these papers as well. We found, in an [interesting new result](https://imgur.com/a/7npD6pO), that our key regression result survives the addition of a cross-model consistency covariate (and in fact the coefficient on self-consistency increased).

---

### Official Review · Reviewer_4uBP · 2026-03-16

**Soundness:** 4
**Presentation:** 4
**Significance:** 4
**Originality:** 4
**Overall Recommendation:** 5
**Confidence:** 2

**Summary:**

This paper introduces the concept of "generator-evaluator self-consistency", which checks if the model applies the same concept (i.e., reusable piece of knowledge such as a rule, category, or principle) during generation and evaluation or its own generation. The paper introduces a well structured formulation and taxonomy in section 2,3, and convinces why this framework is needed through section 6 in a clinical case.

**Compliance With Llm Reviewing Policy:**

Affirmed.

**Key Questions For Authors:**

see weaknesses above

**Limitations:**

yes

**Strengths And Weaknesses:**

* I really liked that fact that the authors included the experiments in Section 6, which extends beyond Section 5. This helped me a lot to understand why we might want to measure self-consistency between generation and evaluation.
* The results in Table 1 and Figure 3 are statistically rigorous, which reports the appropriate statistical measurements to justify the conclusions are valid.
* The scope of the study is wide, in which a lot of follow-up works could be conducted in this direction.

As for weaknesses,
* Overall I have no concerns with this paper, but could the authors provide more guidelines in how people could apply this concept to make agents more reliable (i.e., a solution to improve the metrics that you proposed)?

---

> ### Author Rebuttal · Authors · 2026-03-31
>
> Thank you for your review. We’re glad that you found our work novel, rigorous, and empirically interesting. Below, we respond to your comments.
>
> **We hope our comments have properly addressed your concerns. If not, please let us know if you have any more questions in the follow-up.**
>
> > Suggestions: “Could the authors provide more guidelines in how people could apply this concept to make agents more reliable (i.e., a solution to improve the metrics that you proposed)”
>
> We offer the following clear guidelines for how to use our study to make agents more reliable, both via deployment interventions and training interventions.
> - Deployment: We recommend computing concept-level self-consistency scores across concepts relevant to a particular deployment, across a few models. Flag concept, model pairs with high aggregate self-consistency on a concept, especially when paired with low cross-model consistency – this is the diagnostic signal our regression links to deployment failures (see [new analysis](https://imgur.com/a/7npD6pO) adding cross-model consistency to the deployment regression).
>   - Pre-deployment:
>     - Practitioners can decide not to deploy autonomously on tasks related to flagged concepts, require human oversight for those cases, or select between models based on which has fewer flagged concepts on relevant domains.
>   - Designing pipelines:
>     - A common measure for safety is adding a self-critique step – our results suggest this is unreliable because model’s self check replicates the same omission.
>     - For flagged concepts, pipelines can route to an external evaluator or human reviewer, or practitioners can select among models the one with fewest relevant flags.
> - Training: Flag high self-consistency and low cross-model consistency concepts.
>   - These are concepts where targeted training is most needed. The suite provides this diagnostic automatically and without labels so it can be generated at scale in any domain.
>   - Practitioners can invest annotation effort specifically on flagged concepts rather than on all concepts uniformly.
>   - Additionally, for tests where the generator is instructed to produce a specific flaw (e.g., incomplete reasoning), consistency failures produce concrete training examples where the evaluator missed a known flaw – with labels derived from the test design, not human annotation
>
> Actionability is one of the most important contributions of our paper and have added a section discussing it. Note that the automatic, label-free nature of our metric is what makes these interventions possible and scalable. To clearly highlight the distinguishing advantages of our metric, we also created a [comparison table here](https://imgur.com/a/BBQ7erR).

---

### Decision · Program_Chairs · 2026-04-30

**Decision:**

Accept (regular)

**Comment:**

The paper introduces a concept called generator-evaluator self-consistency to measure if an LLM appears the same underlying concept consistently when switching between generation and evaluation tasks. With experiments on both generic and domain-specific benchmarks (finance, medicine, etc), the paper finds that high internal consistency does not necessarily correlate with accuracy, and may actually indicate a model’s commitment to incorrect or incomplete reasoning. On a medical task, they find that models with higher self-consistency are more likely to commit physician validated mistakes.

Some reviewers pointed out the phenomenon had not yet been isolated with other related phenomena, such as self-endorsement / self-preference bias (models preferring their own generations). In at least one experiment, the same model acts as both the generator and the evaluator. The authors attempted to clarify during the rebuttal period, but the discussion on “inter-model” and “intra-model” between the authors and at least one reviewer should be taken as feedback to increase the clarity of the paper writing. The authors demonstrated during the rebuttal period with new experiments that the within-model consistency metric is a distinct signal different from cross-model consistency. Finally, the findings in the medical setting and the evidence on how the metric is correlated with deployment vulnerabilities was interesting.